# Conservation Studies on Groundwaters' Pollution: Challenges and Perspectives for Stygofauna Communities

Raoul Manenti [1,2,*], Beatrice Piazza [1], Yahui Zhao [3], Emilio Padoa Schioppa [4] and Enrico Lunghi [3,5,6,7]

1   Department of Environmental Science and Policy, Università degli Studi di Milano, via Celoria, 26, 20133 Milano, Italy; betrice.piazza@studenti.unimi.it
2   Laboratorio di Biologia Sotterranea "Enrico Pezzoli", Parco Regionale del Monte Barro, Località Eremo, 23851 Galbiate, Italy
3   Key Laboratory of the Zoological Systematics and Evolution, Institute of Zoology, Chinese Academy of Sciences, Beijing 100101, China; zhaoyh@ioz.ac.cn (Y.Z.); enrico.arti@gmail.com (E.L.)
4   RULE—Research Unit of Landscape Ecology, Department of Earth and Environmental Sciences, University of Milano-Bicocca, Piazza della Scienza 1, 20126 Milano, Italy; emilio.padoaschioppa@unimib.it
5   Museo di Storia Naturale dell'Università degli Studi di Firenze, "La Specola", 50124 Firenze, Italy
6   Natural Oasis, 59100 Prato, Italy
7   Unione Speleologica Calenzano, Calenzano, 50131 Firenze, Italy
*   Correspondence: raoul.manenti@unimi.it

**Abstract:** Assessing the effects of pollution in groundwaters is recently considered among the most relevant aims for subterranean biology; with this perspective, we aim to provide examples of the most relevant effects that pollution may cause on stygofauna community and underline patterns deserving further investigations. We retrieved different cases in which pollution caused alteration of groundwater trophic webs, favored invasions by epigean mesopredators, damaged stygobiont keystone species, and promoted interspecific competition between stygobionts and epigean animals. The results and the remarks derived from our perspective review underline that pollution may play multifaceted effects on groundwaters communities, and the paucity of information that exists on community-level changes and threats underlines the necessity for further studies.

**Keywords:** invertebrates; macrobenthos; sewage; mining; detritus; minerals; stygobite Niphargus; flatworm; aqueduct; salamander; freshwater benthos

## 1. Introduction

Biodiversity conservation is a key aspect for a sustainable management of freshwater ecosystems [1,2]; however, its importance is often overlooked in favor of energy production or water flow control [3]. Biodiversity plays a major role in maintaining freshwater ecosystems functionality [4], but it is threatened by numerous anthropic pressures, such as pollution, habitat alteration, and climate change [5–9]. Freshwater biodiversity protection involves groundwaters, upstream drainage networks, sewage drains locations, riparian structure, and landscape management. All these aspects correlate with both ecosystem health and water quality, determining the assemblages of aquatic species [4,10]. Biodiversity of groundwaters is substantially different in terms of morphological, behavioral, and physiological adaptations from that occurring on the surface [11]. Moreover, groundwaters may also host relict species belonging to phylogenetic lineages that disappeared on the surface [12]. Groundwaters are environments with peculiar hydrogeological features, which are considered among the most fragile natural environments of the world [13]. In these environments is held the biggest reservoir of unfrozen freshwater, thus they represent a fundamental resource for life on Earth [14]. Species that inhabit and only complete their life cycle in groundwaters are called stygobionts (the suffix "stygo" derives from the river Styx, the river flowing in the mythological Greek underworld). They show the highest degree of adaptation to subterranean environments and are often characterized by

peculiar morphological features such as anophthalmia, depigmentation, and elongation of appendages (Pipan and Culver 2012, Romero 2009). Nevertheless, also species that are not strictly linked to groundwater can exploit such environments for a substantial part of their life, being able to reproduce there [15,16]. The facultative cave species that are able to reproduce in subterranean environments are called stygophiles, while those that cannot are the stygoxenes. The facultative cave species can play a fundamental role in shaping stygofauna composition and dynamics [17,18], especially at the borders between groundwaters and surface [19,20].

Pollution is a major threat for freshwater ecosystems; particularly, environmental degradation is depleting water resources, whether because the surface waters contain higher amounts of pollutants, or because pollutants through the soil attain even the underground water reserves [21]. Apart from industrial discharging in large river basins, a huge impact is determined by organic civil pollution that may not only affect small streams and creeks but also groundwaters through percolation, with cascading detrimental effects for humans and for a large part of the overall biodiversity. In general, both shallow groundwater habitats, including hyporheic, and deep aquifers are subject to pollution threats [22,23]. The list of interesting anthropogenic pollutants in groundwaters is a long and growing one.

In both Europe and the USA, the employment of fertilizers and pesticides in agriculture is threatening vast extensions of groundwaters at various depths in both bedrocks and alluvial deposits [24–26]. A widespread driver of pollution is also the occurrence of human and animal wastes; they can provide detrimental levels of organic pollution, especially through uncollected sewages, a threat that was reported from the aquifers of numerous karst areas [27,28]. Even the release of guano by bat colonies can be considered a driver of organic pollution for some subterranean watercourses, as it strongly changes the trophic levels for stygofauna [29]. A number of toxic metals and substances may also reach groundwaters in both karst and not karst areas. Some of them are already well known derivatives from mining activity from direct use of water resources during material washing or linked to the percolating waters in abandoned mines that can spread high contents of heavy metals [30,31]. Some others are emerging contaminants which only recently started to pose threatens for aquifers [13] and, potentially, for their fauna [32]. As an example, there is an increasing pollution of groundwaters derived from pharmaceuticals and personal care products (PPCP), in which compounds can be detected in both urban and rural agricultural areas [13]. Moreover, perfluorinated compounds (PFCs) were reported to impact groundwaters after having moved through soil matrices, as observed in Australia [33], and even aqueous film-forming foams (AFFF) that are typically used in firefighting operations may pollute groundwaters [34].

Assessing the effects of pollution is recently considered among the most relevant aims for subterranean biology [35]. However, the impact of pollution on stygofauna varies according to pollutants typology and abundance [32], and assessing its effects may not be trivial. Even in cases of relatively small concentrations of pollutants in soils and epikarst, their release may be prolonged and determine a chronic groundwater pollution [36] with unexpected detrimental effects. As stygobionts usually have long life cycles [37], the comprehension of exposure to certain compounds is far from being understood and may have implications on survival, fitness, and fertility rates. Stygofauna is likely intolerant to even a small alteration of such chemical concentrations [32]. Assessing the effects of pollution on the communities that inhabit groundwaters is of fundamental importance for the development of proper and effective strategies to contrast it.

With this contribution, we aimed to provide examples of the most relevant effects that pollution may cause on stygofauna communities; the rationale of this perspective was to provide a theoretical and conceptual framework of the processes that may involve pollution in disrupting stygofauna communities, evidencing their generalizability for further experimental studies. Particularly, we discuss how pollution can affect stygobiont

keystone species, whole trophic webs, the ecological niche of epigean mesorpedator species, and how it can alter the interspecific competition between stygobionts and epigean animals.

## 2. Pollutants Affecting Groundwater Animals

Numerous pollutants may act as anthropogenic stressors for stygofauna, as recently highlighted by an extensive review on the subject [32]. These pollutants include pesticides, fertilizers, metal, salt water, different volatile organic compounds (including aromatic compounds such as benzene), and likely also different emerging pollutants such as microplastics and flame retardant tris (1-chloro-2-propyl) phosphate (TCPP) [32,38]. The recent review of Castano-Sanchez, Hose and Reboleira [32] was quite exhaustive, thus it is preferable for readers to refer to it. For the purposes of this perspective, it is important to underline how the effects of these pollutants were observed in both stygophile and stygobiont species by generally measuring in experimental conditions the mortality of field collected individuals [32,39]. Most studies deal with single species, and the effects on the whole groundwater community remain to be assessed; additionally, the choice of the organisms is not mainly based on their functional role they play in the community but on a few abundant and easy-to-sample model species [38,40,41].

## 3. Relevance of Effects of Groundwater Pollution for Stygofauna in Recent Literature

Considering the groundwater community level, we performed a systematic evidence review (SER) to find unbiased data dealing with stygofauna and pollution. We applied the PRISMA (preferred reporting items of systematic reviews and meta-analyses) guidelines [40], and we searched the Web of Science database for peer review papers dealing with both stygofauna and fauna living in spring habitats. The database indexes metadata of scientific literature published between 1965 and 2019. We used a searching string designed to collect all articles in the whole database that could contain information of fauna observations in caves; in May 2020 from Milano (Italy), we used the key words "groundwater fauna" (GF) as the topic. For the search, we used a PC ASUS K501 with the Google Chrome browser, after having emptied its cache box. The screening was performed by one of us (BP). We initially cleaned the dataset by discarding all the articles that contained information that was not related to the study object. We rejected articles about botany, paleontology, geology, and all their sub-disciplines (paleoecology, stratigraphy, geomorphology, etc.), and the articles focused on subterranean environments or groundwater that did not mention animals. Articles concerning single species or taxa not belonging to stygofauna or those lacking hypogean representatives or concerning terrestrial environments, estuaries, swamps, mangroves, streams, rivers, lakes, and all saltwater environments were also discarded. Finally, after the first screening, the same author performed a second selection procedure, in which all the articles that did not contain enough information or were in a language other than English were removed. From the papers, we collected the following information: number of species mentioned, number of surface species found in groundwaters, typology of the study (distinguishing between ecology, taxonomy behavior, conservation, and faunistic assessment and considering that the same paper could belong to multiple categories), and if the paper mentioned or did not mention the occurrence of pollution in its study site/area or in the collection point of the studied animals.

To assess the relationship between the selected document features and the relevance of groundwater pollution, we built a generalized linear model (GLM) with binomial error distribution with the mention or not of pollution as the dependent variable. As fixed factors, we used the paper typologies and the number of surface species mentioned in groundwaters (Equation (1)). Through a likelihood ratio test, we assessed the significance of the fixed factors composing the GLM model [41]. We verified model assumptions by plotting residuals versus fitted values, and we assessed VIF values to verify the absence of multicollinearity issues. All the analyses were performed in the R 3.6.3 environment.

$$\begin{aligned} Pollution \sim\ & Ecology + Taxonomy + Fauna.assessment + Conservation + Behavior \\ & + Number\ of\ surface\ species\ mentioned\ underground\ , family = binomial \end{aligned} \tag{1}$$

Results evidenced that, among the 275 papers dealing with groundwater fauna selected through the systematic evidence review, only 20 reported or studied pollution in groundwaters (Table S1, List S1). The analysis revealed a negative relationship between reporting pollution and taxonomical papers and fauna assessments. Moreover, we detected a weak positive relationship between the occurrence of surface fauna underground and pollution (Table 1).

**Table 1.** Results of the GLM analysis on the relationship between papers on subterranean biology that mention the occurrence of pollution, the typology of the papers, and the presence of surface species underground. In bold are the significant relationships.

| Variable | Estimate | $\chi^2$ | P |
|---|---|---|---|
| Ecological papers | −0.31 | 0.31 | 0.52 ta |
| Taxonomical papers | −18.05 | 17.5 | **<0.01** |
| Fauna assessments | −2.32 | 14.2 | **<0.01** |
| Conservation papers | −0.69 | 0.44 | 0.50 |
| Behavior | −17.8 | 0.95 | 0.32 |
| Surface species underground | 1.42 | 3.61 | **0.05** |

These results underline the low relevance that is given to pollution in studies on groundwater fauna and provide a first indication of the fact that it may cause overlooked patterns of species occurrence in subterranean water systems. We discuss the patterns that pollution may alter at the community level of groundwaters in Section 4.

## 4. Effects and Research Perspectives at Community Level

While a relatively rich literature exists on the effects of pollution at the single species level of typical stygobiont species (see the recent review of [32]), few conservation biology and sustainability studies embrace a large part of the groundwater community [42–46]. A whole perspective is important to perform remediation actions and management activities that could be effective at the scale of whole groundwater systems impacted by pollution.

Multiple factors are reported to shape species composition of communities occurring in a groundwater system [47], with no single factor providing a complete explanation for the observed patterns, even in the relatively simple conditions offered by subterranean environments [23]. The complex interactions between species' physiological constraints, evolutionary processes, and selective pressures currently acting in both surface and subterranean adjacent environments shape the proportion between species stygobionts and species typical of epigean freshwaters that compose the groundwater community (Figure 1). In natural conditions (Figure 1B), species composition of a local groundwater community is a consequence of multiple factors interacting in a hierarchical fashion. Along with the evolutionary and the historical events that shaped stygobionts' adaptations, chemophysiological constraints limit the interchange between epigean and subterranean species. Pollution may influence contemporarily more of these factors, disrupting the community features at different levels. In particular, when pollution occurs (Figure 1C), it may alter interspecific interactions, epigean species' dispersal ability, and habitat selection with deep consequences at the whole community level. We can recognize two main different typologies of effects. First, we can distinguish the effects that act at the core level of groundwater communities represented by stygobionts [48]. Pollution may affect stygobiont keystone species and alter stygobiont functional roles in the trophic webs. Secondly, it is possible to recognize effects that act at the border level between groundwaters and surface freshwater, potentially affecting groundwater colonization by surface species. We discuss challenges and implications of these two effects in the next paragraphs.

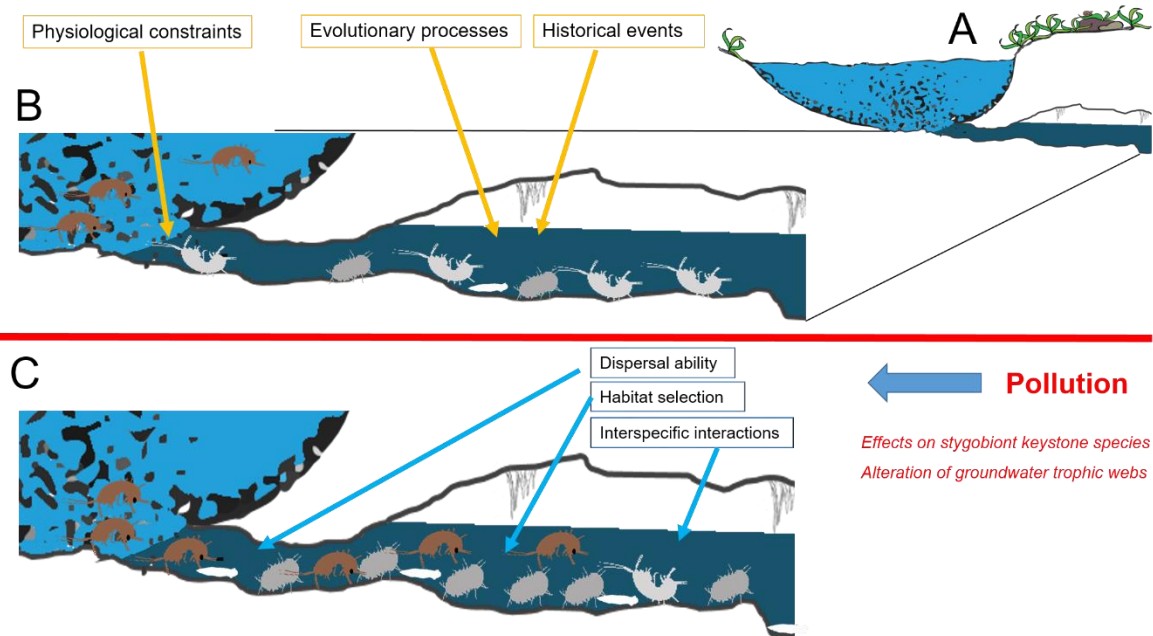

**Figure 1.** Diagram of the terminal stretch of a groundwater system (**A**). In natural conditions (**B**), species composition of a local groundwater community is a consequence of multiple factors interacting in a hierarchical fashion. Along the evolutionary and the historical events that shaped stygobionts' adaptations, chemo-physiological constraints limit the interchange between epigean and subterranean species. When pollution occurs (**C**), effects on stygobiont keystone species and trophic webs may alter interspecific interactions, epigean species' dispersal ability, and habitat selection with deep consequences at the whole community level. Brown figures exemplify epigean animals (amphipods), while grey and white ones represent stygobionts (amphipods, isopods, and planarians).

### 4.1. Effects on Stygobiont Keystone Species

Stygobiont vertebrates usually are the top predators of the groundwaters they inhabit. As top predators, they likely play keystone roles for the groundwaters communities [49], even if it is difficult to find studies dealing with top-down effects played by top predator stygobiont vertebrates in groundwaters. Among vertebrates, two groups that successfully entered and adapted to subterranean aquatic environments are cave salamanders and cavefishes; direct and indirect information on the possible effects of pollution occur for some of them.

In Europe, the olm (*Proteus anguinus*) is the most well-known species of stygobiont vertebrate and one of the most outstanding cave dwellers of the karst. Some past studies measured metal concentrations in the olms' tissues, providing comparisons with the concentrations occurring in the surrounding groundwaters [50,51]. Although groundwaters were not polluted with metals, olms showed higher concentrations of mercury, zinc, and copper, especially in the liver; individuals of the subspecies *P. anguinus parkelj*, that are locally linked to some spring habitats showed also higher levels of arsenic, especially in the integument [50]. Considering the long life span of the species [52], these data suggest that olms can accumulate metals and toxic substances with potential negative consequences on their fertility, fitness, and survival [51]. Considering that even animals with short lifespan can accumulate contaminants, the effect can be enhanced in stygobionts with long life cycles. Bioaccumlation may be particularly relevant for top predators, even if pollution is likely to affect all the levels of the trophic web. Assessments of the role played by olms' occurrence and density on whole groundwaters' communities could be particularly important to highlight the key role that stygobiont salamanders may play and to understand how pollution may disrupt it.

When predator vertebrates are not present, other stygobiont species may occupy top predator level and be affected by pollution with potential cascade effects for the whole

community. This could be the case of stygobiont planarians; they are predator flatworms with specialized structures and behaviors that allow them to prey upon all the other aquatic invertebrates [53]. Thus, stygobiont planarians often occupy the highest position of the trophic webs of small subterranean river courses and groundwater interstices [54,55]. Even if there are no direct studies on the effect of pollution for stygobiont planarians, recent research underlined that 30% of the known sites of all the stygobiont planarians of Northern Italy shows signs of pollution, which likely determines the disappearances of species locally [56]. Most genera of epigean planarians are particularly sensitive to organic matter and chemical pollution (Reynoldson and Young, 2000), therefore, it is possible that groundwater pollution has similar detrimental effects on the underground species of the same genera.

The effects of planarians disappearance in polluted groundwaters still need to be assessed and understood, but it is likely that cave-dwelling triclads play control roles for other stygobiont and stygophile invertebrates, mainly crustaceans, bivalves, and annelids, which are their main prey [57].

### 4.2. Alteration of Groundwater Trophic Webs

Several metals and contaminants were reported to cause mortality in a number of stygobiont species that do not necessarily occupy key positions in the groundwater trophic web [32]. The effects of such contaminants can vary according to the duration and the intensity of the pollution events. In the case of chronical events with extensive releases of toxic contaminants, the effects on the groundwater trophic webs are likely to resemble those observed in epigean freshwaters, where strongly unbalanced communities are characterized by a few dominant, tolerant species, usually detritivores [58,59]. Unfortunately, despite several anecdotal reports of subterranean rivers and streams being chronically or occasionally polluted by heavy metals and xenobiotic compounds, no studies detail the features of their stygofauna; as an example, hearsay affirms that, because of chemical pollution, the stygofauna of one of the most important Italian subterranean rivers, the Timavo, was extremely impoverished during the 19th century, with the disappearance of both top predators such as the olm and typical detrtivore species such as the shrimps of the genus *Troglocaris*. Studies depicting such effects and providing management perspectives in terms of recovery of the stygofauna communities can be particularly important, especially for the karst areas occurring in emerging countries where pollution effects can be overlooked by current policies.

Something much more challenging is addressing the effects of organic pollution. In temperate regions, groundwaters are usually oligotrophic environments, especially if compared to the surface water bodies [60]. When organic pollution occurs, major changes in the trophic web structure are likely [61]. The first strong effect is on biofilms, which strongly increases their shape and biomass [62]. Thus, signs of organic enrichment in subterranean streams and pools can be easily detected when the sites are accessible [56]. Both organic compounds and biofilms may constitute a resource for stygobionts but may also be detrimental. Indeed, in a review of the few studies documenting the effects of organic pollution on cave macrobenthos, Wood et al. [63] showed that responses at the community level may be quite variable. Particularly, not only reductions in abundance or local extinctions of stygobionts were reported but also cases of increase in the abundance of certain species and trophic groups. Among taxa that could be favored by nutrient enrichment, there are stygobiont isopods. They usually occupy the role of primary consumers and feed on decaying organic material [64] and biofilms [61]. Different past studies evidenced high abundances of isopods in polluted subterranean streams. For example, Holsinger [65], in a cave polluted by sewage, observed in the same sampling occasions both the stygobiont isopod *Caecidotea recurvata* and its predator *Phagocata subterranea*. Maximum density of isopods was particularly high (60.9 individuals/m$^2$) compared to that that of planarians (25 individuals/m$^2$) [65]. More than 30 years after, a study performed on the same site that considered only crustaceans revealed that the maximum density of *C. recurvata* was even

higher in moderately and slightly polluted pools (74.6 individuals/m$^2$), while the stygobiont *C. recurvata* was only present in unpolluted water bodies of the cave [66]. Even though there is no information on the situation prior to pollution, organic enrichment seems to have privileged only one species of primary consumer that became dominant. Moreover, the main predator of this primary consumer was apparently favored, while its predator other species with different tolerance and possibly trophic niches disappeared [63,66]. A similar situation with dearth of amphipods and richness of isopods following organic pollution was recorded by Graening and Brown [29]. In that case, the alteration of the groundwater trophic web apparently provided advantage to a cavefish predator, *Amblyopsis rosae*, and diminished the occurrence of a salamander predator, *Typhlotriton spelaeus* [29].

A natural way of organic enrichment for groundwaters is represented by bat guano that may strongly affect groundwater's community structure [63]. Bat guano not only may supply sufficient resources to decrease the selective pressure posed by oligotrophy, but when it is particularly abundant, it may also generate situations similar to anthropogenic organic inputs for stygofauna. A case of natural organic input suggested to be detrimental is represented by *Paradactylodon gorganensis*, a salamander that is considered as a valid species as an ecotype of *Paradactylodon persicus*. *P. gorganensis* breeds in the terminal trait of an Iranian subterranean stream, which seems polluted by the guano of the bat *Myotis blythii* [67], posing questions of whether to limit bat occurrence in that site. In this case, further studies are necessary to assess if the organic enrichment could be a threat or, instead, provides resources for the prey of the salamanders.

Moreover, as guano abundance reflects bats' populations trends, it may be a variable resource. Currently, bat populations are declining, and this decline may be detrimental also for groundwaters communities that rely on guano for their survival [68]. Assessing how bat populations trends result in changes in groundwaters' community structure could be a promising field of research as the hypothesis that minor organic enrichment/pollution may be advantageous to stygofauna under some circumstances [45].

*4.3. Effects on Interspecific Competition between Stygobionts and Epigean Animals*

One aspect that often strongly limits the advantages of groundwaters' organic pollution for stygofauna is that it often allows epigean species to successfully invade subterranean habitats [63]. The enrichment of trophic resources is likely to reduce the constraints that limit epigean taxa dispersion in subterranean environments. The reduced metabolism that characterizes cave organisms and their relatively low rates of activity if compared with surface species are considered advantageous in oligotrophic waters [69,70]. However, when trophic resources increase, the higher metabolic rates in stygophile species may allow them to outcompete stygobionts [70]. We recorded a similar situation in the cave "Grotta di Bocca Lupara" in Liguria (North-western Italy) during a survey performed from December 2017 to March 2018. The subterranean stream flowing through the cave showed strong cover of biofilm/periphyton and evident signs of pollution linked to bat guano and likely to uncollected sewages. Along the stream, we recorded, over different transects, high abundances of the epigean gammarid *Echinogammarus* sp. gr. *veneris* (maximum abundance: 8.64 individuals/m$^2$), most of which were depigmented, suggesting a possible ongoing adaptation to the subterranean environment (maximum observed abundance of depigmented individuals: 6.57 individuals/m$^2$). Only few individuals of the stygobiont *Niphargus speziae*, for which the cave is the type locality, were observed (maximum abundance 0.36 individuals/m$^2$). Other cases of epigean gammarids colonizing groundwaters after being polluted were reported for the Peak Speedwell Cavern system (Derbyshire, UK), where *Gammarus pulex* invaded subterranean habitats where it was not previously recorded [63]. When groundwaters reach high levels of pollution, other epigean *taxa* that are tolerant in surface waters may prevail; especially Tubificidae and other annellids are reported to successfully colonize highly polluted subterranean streams [46,71].

*4.4. Effects on Ecological Niche of Epigean Mesorpedator Species*

Mesopredators are predator organisms that do not occupy apex positions of the food web but exhibit predatory behavior to gather trophic resources. Generally, the condition of being a mesoporedator or a top predator is strongly context dependent and may differ between habitats and during ontogeny [72]. In surface streams, a classical example is played by dragonfly larvae; when they are newborn, they occupy an intermediate position in the food webs, and once they grow, they may become the apex predators depending on whether fish are present [73–75]. Caves, especially in temperate areas, show strong differences in abundance and in diversity of both predator and trophic resources if compared to superficial habitats [76,77]. Caves, even if the availability of prey is reduced, can be safe habitats for typical surface mesorpedator species that become the apex predators there [78,79]. How can groundwater pollution promote the shift of an epigean mesopredator into a subterranean top predator? If chemical pollutants are generally depriving most groundwater fauna, reducing the prey available even for tolerant predators [29], organic pollution may provide favorable conditions to epigean mesopredators. The increased abundance of potential prey that we described in the previous paragraphs can enhance the probability of exploitation by surface predators [80]. As an example, in the above mentioned cave "Grotta di Bocca Lupara", we detected at each survey a typical epigean predator species, the leech *Herpobdella testacea*, that was relatively abundant (maximum abundance 2.26 individuals/m$^2$). This leech is tolerant to pollution [81] and resulted as the top predator of the investigated subterranean stream. Similarly, different cave-dwelling populations of typical surface leech species are reported for caves that apparently show high levels of trophic resources and prey [82,83], even if this correlation still needs to be tested. Another example is that of fire salamander larvae that, when in streams, are typical mesopredators facing the risk of fish and dragonfly larvae, but when larvae are laid in caves by their mothers, they are at the top of the freshwater food chain, and the only predation risk that they face is at the intraspecific level [84]. Fire salamander larvae occurrence in subterranean aquatic environments is positively correlated with the abundance of potential invertebrate prey [85].

## 5. Conclusions

Groundwaters, as with other subterranean environments, are under different threats that need urgent contrasting actions to preserve the stygofauna communities they host. Groundwaters are also useful environments to study ecological and evolutionary consequences of anthropogenic changes and develop remediation actions at multiple spatial scales. This perspective provides different examples of the most relevant effects that pollution may cause on groundwater communities. These examples suggest that, to address pollution threats, it may be useful to view subterranean environments as cybernetic systems which can be studied in light of the energy transfers that occur within and between them as well as in adjacent ecosystems [47]. Pollution events may alter energy flows both directly and indirectly depending on timing and typology of pollutants. In this review, we synthetized how these changes are reflected by changes in the groundwaters' communities, which in turn may reveal fundamental insights to understand the processes involved in adaptation to subterranean environments and to contrast anthropogenic changes. However, the concluding remark deals with the difficulty that exists in understanding community-level changes and threats, because little information exists on groundwaters communities. Studies on stygofauna are often focused on single specialized taxa [86] and rarely consider the role of epigean species entering groundwaters [17]. Paucity of information makes it difficult to compare pre- and post-pollution conditions and know which were the community pre-disturbance features. With this paper, we hope to underline the fragility and the vulnerability of groundwaters and to stimulate future studies on the impact of pollution upon aquatic cave communities and its cascading effects on cave and springs trophic webs.

**Supplementary Materials:** The following are available online at https://www.mdpi.com/article/10.3390/su13137030/s1, Table S1: Data gathered with the systematic evidence reviewon the Web of Science database using the key words "groundwater fauna" (GF) as the topic. List S1: complete reference list of the references used in Table S1.

**Author Contributions:** Conceptualization and first draft, R.M.; literature systematic review, B.P., writing—review and editing E.L., Y.Z., E.P.S. and R.M. All authors have read and agreed to the published version of the manuscript.

**Funding:** This research was funded by THE MOHAMED BIN ZAYED SPECIES CONSERVATION FUND, grants numbers: 162514520; 180520056.

**Institutional Review Board Statement:** Not applicable.

**Informed Consent Statement:** Not applicable.

**Data Availability Statement:** Data and papers used for this perspective review are available upon request.

**Acknowledgments:** We are grateful to Andrea Melotto and Bendetta Barzaghi who helped us with graphics and suggestions; we thank Fabio Stoch for information on the taxonomic status of macroinvertebrates of the cave "Grotta di Bocca Lupara". E. Lunghi is supported by the Chinese Academy of Sciences President's International Fellowship Initiative for postdoctoral researchers (2019PB0143).

**Conflicts of Interest:** The authors declare no conflict of interest. The funders had no role in the design of the study; in the collection, analyses, or interpretation of data; in the writing of the manuscript, or in the decision to publish the results.

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
