# Peer review of "Conservation Studies on Groundwaters’ Pollution: Challenges and Perspectives for Stygofauna Communities"

_sustainability, doi:10.3390/su13137030_

Round 1
Reviewer 1 Report
The review concerns conservation studies on groundwaters’ pollution: challenges and perspectives for stygofauna communities. The topic is certainly interesting and within the scope of the journal. However, some substantial concerns about the paper are as follows: The academic significance of the current study could be clearly stated in the Introduction section. The manuscript does not answer the question raised in its abstract/introduction (lines 28-29). I did not find clear information in the conclusion for stimulating future studies on the impact of pollution upon aquatic cave communities and its cascading effects on cave and springs trophic webs. I recommend to the Authors change the general idea of the manuscript, or clearly state the answer in the Conclusion section. This must be clarified. Articles do not indicate a detailed discrepancy in published research results. Are publications indexed in Wos/Scopus?
To clarify some aspects, I would suggest that the authors write the bibliography evenly: the abbreviation, the DOI number.
Author Response
We are glad for the possibility to revise the manuscript and we are grateful for the precious and detailed comments provided by the reviewer. We provide here below a point by point list of the changes performed thanks to the suggestions provided that allowed us to increase the quality of the manuscript.
The review concerns conservation studies on groundwaters’ pollution: challenges and perspectives for stygofauna communities. The topic is certainly interesting and within the scope of the journal. However, some substantial concerns about the paper are as follows: The academic significance of the current study could be clearly stated in the Introduction section. The manuscript does not answer the question raised in its abstract/introduction (lines 28-29). I did not find clear information in the conclusion for stimulating future studies on the impact of pollution upon aquatic cave communities and its cascading effects on cave and springs trophic webs. I recommend to the Authors change the general idea of the manuscript, or clearly state the answer in the Conclusion section. This must be clarified
Following the suggestions, we modified extensively the text in the abstract and in both introduction and conclusion sections in order to be more adherent to the results achieved and to better evidence the rationale of the paper. According to the comment we down-toned the general idea of the manuscript and we better focussed on the results obtained and we clarified which aspects could be more in depth investigated. In that way we avoided the readers to create too much expectatiosn that our perspective could not satisfy , but we stressed for the aspects that miss generally in the literature and that could be investigated in the future.
. Articles do not indicate a detailed discrepancy in published research results. Are publications indexed in Wos/Scopus?
Yes papers are indexed; we specified it now following the suggestion at lines 117-119. Following also the next cmment we now provide both the completetavble of the papers used for the first chapter analysis in which we performed a systematic evidence review, both the complete reference list.
To clarify some aspects, I would suggest that the authors write the bibliography evenly: the abbreviation, the DOI number.
To clarify the aspects explained by the reviewer, we provided the references used for the systematic evidence review; particularly we provide the table of the data as Supplementary Table 1 and the list of the references used as another Supplementary Appendix; we list all the other cited references in the references list. For both references lists we used the MDPI style to improve clarity.

Reviewer 2 Report
In this manuscript, the authors conducted a review regarding challenges and perspectives for stygofauna communities through a lens of groundwater pollution. The following concerns shall be addressed.
- It would be ideal to have one section for pollutants, in which the authors shall review the impact of each pollutant. Also saltwater intrusion shall be taken into account.
- In my opinion, one single section regarding the case of China seems to be out of the context. I would not be surprised to see the case elsewhere apropos of karst areas.
Author Response
We thank the reviewer for the precious and detailed comments. We modified the text following all the suggestions that allowed us to improve the quality of the manuscript.
- It would be ideal to have one section for pollutants, in which the authors shall review the impact of each pollutant. Also saltwater intrusion shall be taken into account.
We added a small section for pollutants. We underline that they have been already and very recently revised by the paper of
Castano-Sanchez et al., 2020 -Ecotoxicological effects of anthropogenic stressors in subterranean organisms: A review. Chemosphere 244:125422.-
- In my opinion, one single section regarding the case of China seems to be out of the context. I would not be surprised to see the case elsewhere apropos of karst areas.
Following the suggestion, we deleted the section on China and we modified the text trying to improve clarity and the comprehension of the whole narrative.
Round 2
Reviewer 1 Report
Accept in present form.
Reviewer 2 Report
I am very happy that the authors have addressed my concerns point by point precisely. No further comments from my side.